There are amendments to this paper

# Hydrophobic gating in BK channels

Zhiguang Jia[1], Mahdieh Yazdani[1], Guohui Zhang[2], Jianmin Cui[2] & Jianhan Chen [1,3]

The gating mechanism of transmembrane ion channels is crucial for understanding how these proteins control ion flow across membranes in various physiological processes. Big potassium (BK) channels are particularly interesting with large single-channel conductance and dual regulation by membrane voltage and intracellular $Ca^{2+}$. Recent atomistic structures of BK channels failed to identify structural features that could physically block the ion flow in the closed state. Here, we show that gating of BK channels does not seem to require a physical gate. Instead, changes in the pore shape and surface hydrophobicity in the $Ca^{2+}$-free state allow the channel to readily undergo hydrophobic dewetting transitions, giving rise to a large free energy barrier for $K^+$ permeation. Importantly, the dry pore remains physically open and is readily accessible to quaternary ammonium channel blockers. The hydrophobic gating mechanism is also consistent with scanning mutagenesis studies showing that modulation of pore hydrophobicity is correlated with activation properties.

[1] Department of Chemistry, University of Massachusetts, Amherst, MA 01003, USA. [2] Department of Biomedical Engineering, Center for the Investigation of Membrane Excitability Disorders, Cardiac Bioelectricity and Arrhythmia Center, Washington University, St Louis, MO 63130, USA. [3] Department of Biochemistry and Molecular Biology, University of Massachusetts, Amherst, MA 01003, USA. Correspondence and requests for materials should be addressed to J.C. (email: jianhanc@umass.edu)

on channels facilitate the flow of ions through cell membranes via opening of their pores in response to various electrical and chemical signals[1]. The transmembrane pore in ion channels generally contains structures that control the types of permeating ions, known as the selectivity filter, and those that open and close to control the flow of ions, known as the gate. Upon sensing physiological stimulations, the channel proteins change conformation to open the gate and allow ion passage. The large conductance $Ca^{2+}$-activated $K^+$ (big potassium (BK)) channels (Slo1) are widely distributed in many cell types[2, 3], play important roles in numerous physiological processes[4, 5], and are associated with pathogenesis of a wide range of diseases including hypertension, autism, epilepsy, stroke, and asthma[6]. BK channels are great model systems for understanding gating mechanisms of ion channels. They have a large single-channel $K^+$ conductance (~100–300 pS)[7, 8], and can be synergistically and independently opened by depolarization of membrane potential and the binding of intracellular $Ca^{2+}$[9, 10]. Intracellular $Mg^{2+}$ also activates the channel by interacting with the voltage sensor[11]. Functional BK channels are homo-tetramers of Slo1 subunits[12], each of which contains three structural domains with distinct functions (Fig. 1a). The voltage sensor domain (VSD) detects changes in membrane potential; the pore-gate domain (PGD) controls ion selectivity and $K^+$ permeation, and the cytosolic tail domain (CTD) senses various intracellular ligands including $Ca^{2+}$. VSD and PGD together form the transmembrane domain (TMD) of a BK channel, consisting of seven membrane-spanning helices, denoted S0–S6. Similar to other 6TM/P potassium channels, S0–S4 and S5–S6 helices form VSD and PGD, respectively. PGD also contains an additional half loop known as the pore (P) loop, which forms the $K^+$ selectivity filter and is highly conserved among potassium channels. Most ion channels with the pore adopting a similar membrane topology feature a cytosolic gate that is formed by the bundle crossing of four S6 helices[13, 14], which creates a physical barrier and blocks the ion permeation pathway in the closed state. However, in BK channels it has been questioned whether a similar physical cytosolic gate exists. The large inner pore of BK channels is accessible to moderately sized quaternary ammoniums (QAs) regardless of the activation states of the gate[15, 16]. Methanethiosulfonate reagents can also access and modify cysteines in the central cavity below the selectivity filter even in the closed state[17]. As such, the current hypothesis is that the activation gate in BK channels lies within the selectivity filter itself[15–19].

A major breakthrough has been made with recent determination of atomistic Cryo-EM structures for the full−length *Aplysia californica* BK (acBK) channel in both metal-bound and metal-free states[20, 21]. These long-awaited structures provide a basis for dissecting molecular mechanisms of ion conductance and sensor-pore coupling in BK channels. Intriguingly, the pore below the selectivity filter is only slightly narrower and remains widely open even in the absence of $Ca^{2+}/Mg^{2+}$ binding (which is expected to represent the closed state) (Fig. 1b). While it is possible that the pore may undergo further closure from the Cryo-EM structures, a physically open pore in the metal-free state is consistent with the observation that the deep-pore region remains accessible to large organic molecules. On the other hand, the structures do not provide any obvious clue to support the selectivity filter gate hypothesis. The structures of the selectivity filter are essentially identical in metal-bound and metal-free states with an RMSD < 1.0 Å[20, 21]. As such, the efforts to locate a physical gate in BK channels have so far failed to identify a credible candidate.

In this work, we show that a physical gate does not seem to be required for closing BK channels. Leveraging efficient graphics processing units-enabled atomistic molecular dynamics (MD) simulations, we suggest that the deep-pore region of the human

BK (hBK) channel (approximately, G310–P320; Fig. 1b) readily undergoes hydrophobic dewetting transitions in the metal-free state, leading to a high energy barrier that prevents permeation of both water and small ions. Initiation of dewetting transitions requires the movement of pore-lining S6 helices in the metal-free state as observed in Cryo-EM structures, which modifies the pore geometry and surface hydrophobicity. Importantly, the dry pore remains physically open; the barrier to $K^+$ permeation arises directly from the vapor phase that separates the selectivity filter from the bulk solution. Free-energy calculations further demonstrate that hydrophobic ions such as small QAs can readily access the deep-pore region with minimal or modest barriers in both dry (closed) and hydrated (open) states. Such a hydrophobic gating mechanism is also consistent with findings of scanning mutagenesis studies showing that modulation of the polarity and hydrophobicity of pore-lining residues can be directly correlated with hBK channel activation probabilities[22]. Additional simulations of selected mutants confirm that the observed change of the channel activation property may be attributed to different dewetting tendency of the mutated pore. We note that the gate observed in this work for the hBK channel belong to an expanding list of examples of ion channels with true hydrophobic gates[23–26], where the barrier to ion permeation arises directly from dewetting transitions instead of physical blockage generated by the subsequent structural collapse of the pore.

## Results

**Metal-unbinding primes hBK pores for hydrophobic dewetting**. Close inspection of the pore reveals relatively subtle, but potentially important structural changes upon metal removal besides modest narrowing of the overall opening profile by −1 to 4 Å at various locations as previously noted[20, 21] (also see Fig. 1b). These conformational changes mainly involve an amphipathic segment of helix S6, $V_{319}PEIIE_{324}$. The S6–S6 helix packing in the open state is mediated by hydrophobic residues such as V319 and I323, which pack tightly with a hydrophobic patch around A313 of the neighboring S6 helix (Fig. 1c). Movement of S6 helices upon metal removal (indicated by the red arrow in Fig. 1c) disengages these hydrophobic contacts, and exposes both V319 and I323 to the pore inner surface (Fig. 1c, d). The S6 movement also effectively rotates conserved E321 and E324, which line the entrance of the pore in the open state (Fig. 1d), to membrane interface-facing orientations. The total water exposed hydrophobic surface area of the pore region increases from ~$1.3 \times 10^5$ to ~$2.1 \times 10^5$ Å$^2$ in the metal-free state. As such, the pore becomes not only narrower, but also more elongated and more hydrophobic upon metal removal (Fig. 1d). These changes can have important implications in gating of BK channels. It has been shown that a hydrophobic pore with ~10 Å or smaller diameter can readily undergo dewetting transitions and become depleted of liquid water, giving rise to an effective free energy barrier to ion permeation[27, 28]. Hydrophobic dewetting has been suggested to underlie the deactivation of several ion channels to facilitate hydrophobic gating and/or structural collapsing, albeit with narrower pores[23–26, 29, 30]. Atomistic simulations have also predicted that ion depletion and dewetting transitions drive the eventual collapse of the pore in voltage deactivation of Kv1.2/Kv2.1 $K^+$ channels[30, 31]. It is plausible that the observed changes in the pore geometry and surface hydrophobicity in metal-free Cryo-EM structures prime the pore for dewetting transitions and represent initial steps toward deactivation of BK channels.

**hBK channels readily dewet in the metal-free state**. Atomistic simulations were performed in explicit solvent for both the metal-bound and metal-free states for up to 800 ns to evaluate the

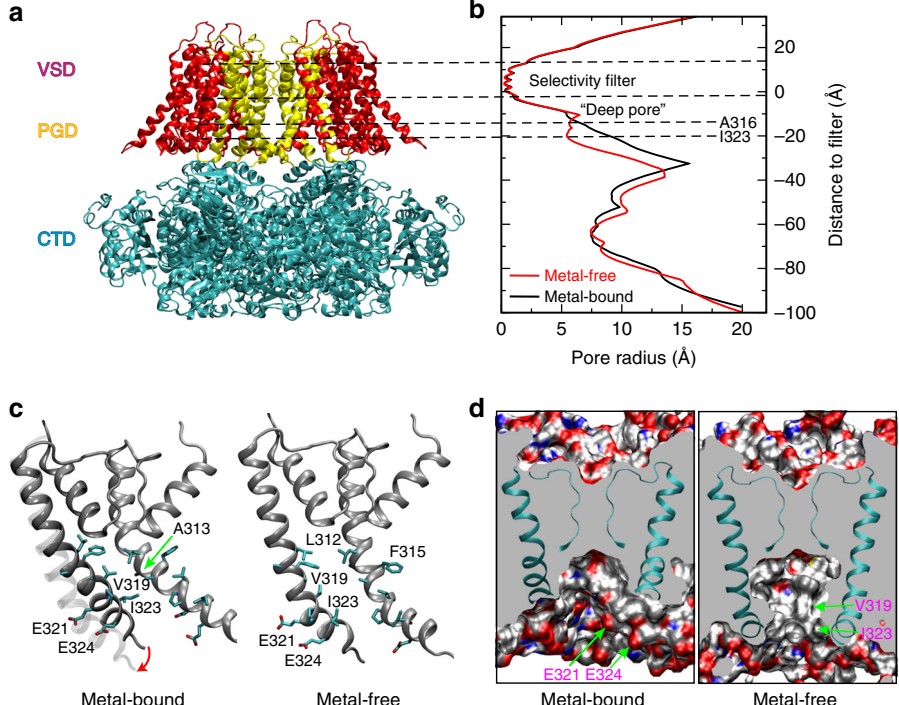

**Fig. 1** Key structural features of the hBK channel. **a** A homology model of the hBK channel in the metal-bound (open) state derived from the Cryo-EM structure of acBK. **b** Pore profiles in the metal-free (red) and metal-bound (open) states, calculated using program HOLE[56] The center-of-mass of the backbone of residues 286–288 was selected to represent the location of the filter entrance (distance = 0). **c** Packing of two neighboring S6 helices in the metal-free and metal-bound states with key residues shown in licorice. The location of the S6 helix in the metal-free state is shown as a transparent trace in the metal-bound structure to illustrate the movement upon metal removal. **d** Longitudinal sections through the center of TMD, showing water exposed surfaces of the pore cavity. Charged oxygen atoms are colored in red, nitrogen atoms in blue, and nonpolar hydrocarbons in silver. Key charged (E321 and E324) and hydrophobic (V319 and I323) residues with drastically different pore exposure in two states are marked for one of the subunits. Parts of the selectivity filter and S6 helices are also shown in cartoon. Lipid molecules within 10 Å from S6 helices were included in the calculation of the water exposed surface

hydration properties of the pore (see Methods). The results are summarized in Fig. 2 and Supplementary Figs. 1, 2. The pore was highly stable and remained fully solvated throughout all three 800-ns simulations of the metal-bound hBK channel, with the number of pore water molecules fluctuating steadily around 40 (Supplementary Fig. 2, traces o1–o3). The backbone RMSD values from the initial homology modeled structure remain largely below 5 Å, with the pore itself deviating no greater than 2 Å (Supplementary Fig. 1). Neither the packing of S6 helices nor the side-chain orientation showed any significant movements during simulation (Supplementary Figs. 3 and 4), and the pore profile remained essentially identical to that of the initial homology model derived from the Cryo-EM structures (Supplementary Fig. 5). In contrast, the pore readily underwent dewetting transitions in the metal-free state, becoming completely depleted of liquid water in six out of eight simulations (Supplementary Fig. 2, traces c1–c8). The dry pore was observed within 200 ns for several runs (c3, c4, c6, and c7). Dewetting transitions usually go through semihydrated states, where the number of pore water molecules drop to around 20 (e.g., see Fig. 2b). These semihydrated states could persist ten to several hundred ns until relatively rapid dehydration transitions to reach dry states that are largely free of liquid water (Fig. 2c). For runs c2 and c5, the pore remained semihydrated even after up to 800 ns. In runs c1 and c7, the pore fluctuated rapidly between fully dehydrated and semihydrated states and did not appear to reach a stable dry state after up to 800 ns. We note that the selectivity filter itself was extremely stable in all simulations, deviating only about 0.5 Å from the initial structures in both metal-bound and metal-free states

(Supplementary Fig. 1e, f). The filter remained fully occupied with $K^+$ throughout these simulations, even though $K^+$ at S0 and S4 sites could undergo dynamic exchanges. Control simulations of the acBK channel show that it can also undergo similar dewetting transitions in the metal-free state (Supplementary Fig. 6a), suggesting that dewetting is likely a conserved property of BK channels. Another control simulation was performed for the metal-free hBK channel using a different force-field/water model combination (Gromos/SPC vs. CHARMM36m/TIP3P), and the result shows that the observed dewetting transition appears robust (Supplementary Fig. 6b). Together, these simulations strongly support that changes in the pore geometry and hydrophobicity upon metal removal as observed in the Cryo-EM structures do prime BK channels to readily undergo dewetting transitions, reaching a stable dry state on the timescale of hundreds of ns to multi μs.

The relatively long timescale required to reach the stable dry state is likely due to additional conformational transitions of the pore associated with this transition. It has been demonstrated that the probability of dewetting is highly sensitive to the pore size and hydrophobicity and that fully and semihydrophobic model nanopores can adopt stable dry states when the radius is less than ~7 and ~5 Å, respectively[29]. Indeed, dewetting transitions of the metal-free state of hBK channels are accompanied with further narrowing of the pore. As shown in Fig. 3a, the dry pore has a radius of ~3–4 Å around L312–P320, compared to that of ~6–7 Å in the initial (homology modeled) structure. The reduction of pore radius is mainly due to further inward movements of S6 helices (Fig. 3 and Supplementary Fig. 3). Such

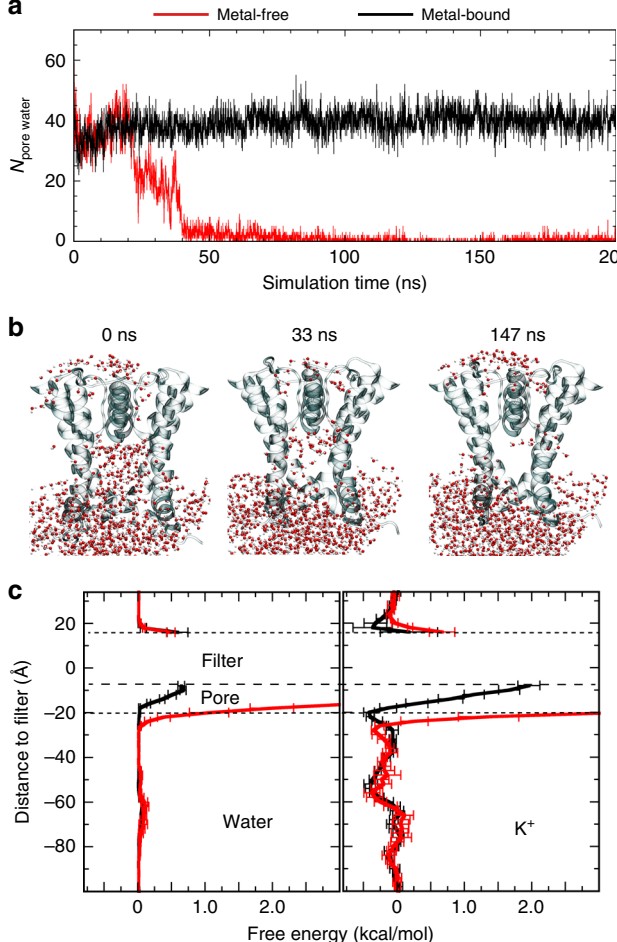

**Fig. 2** Dewetting transitions of the pore of hBK channels. **a** Number of pore water molecules as a function of time from two representative simulations of the metal-free (run c3, red trace) and metal-bound (o1, black trace) states. See Methods for the definition of pore water molecules. **b** Snapshots at three time points illustrating the dewetting transition from simulation c3. **c** Effective free energy profiles of water and $K^+$ derived from the average densities from the last 50 ns of runs o1–o3 (metal-bound) and c3, c4, c6, and c8 (metal-free). Error bars shown are standard errors among runs

(Supplementary Fig. 3c), which contribute to maintaining a hydrophobic deep-pore surface.

**Dewetting transitions give rise to a hydrophobic gate.** Small ions seem clearly excluded from the pore with liquid water upon dewetting transitions; there is virtually no density of $K^+$ or $Cl^-$ observed in the dry pore during equilibrium simulations (Fig. 2c). Umbrella sampling was performed to further quantify the free energy barrier of $K^+$ permeation through the dry pore in the metal-free state. The results, shown in Fig. 4a, confirm that the dewetted region does give rise to large free energy barriers, in excess of 8 kcal/mol, before $K^+$ ions could reach the selectivity filter. This is in contrast to the hydrated metal-bound state of the hBK channel where there is virtually no free energy barrier for $K^+$ to access the deep-pore region (~10–15 Å below the filter). Therefore, dewetting transitions observed in simulations of the metal-free hBK channel represent highly plausible gating events and the channel with a dry pore could likely be the closed state. Importantly, the dry pore remains physically open with an average diameter of ~6 Å. This suggests that in BK channels the dewetted vapor gap directly serves as the activation gate. In other words, hBK channels likely process a true hydrophobic gate that relies on dewetting transitions directly instead of physical blockage for regulating ion permeation. Similar vapor barriers have also been suggested to be involved in gating of other channels[23–26].

A physical open pore in the closed state of BK channels also provides a straightforward explanation for the intriguing observation that moderately sized QA compounds can access the deep-pore region to block the channel even in the closed state during membrane repolarization[15, 16]. Indeed, free energy profiles of pore permeation calculated from umbrella sampling, summarized in Fig. 4b, c, confirm that small QAs such as tetraethylammonium (TEA) and tetrabutylammonium (TBA) can access the deep-pore region of the channel with virtually no barrier in the hydrated open state and minimal barriers in the dry closed state (Fig. 4b, c). The lack of any significant barriers is also consistent with fast blocking and unblocking kinetics in BK channels[15, 16]. Interestingly, the larger and more hydrophobic TBA experiences a lower free energy barrier entering the pore, which is due to more favorable interactions with the hydrophobic inner surface of the pore. Once in the deep-pore region, TEA and TBA have extensive contacts with pore-lining nonpolar residues (Supplementary Fig. 8). The larger TBA in particular is quite stable in the deep pore (Fig. 4b), consistent with its nature as a channel blocker.

**Effects of pore mutations correlate with dewetting tendency.** Three nonpolar residues in the deep-pore region, L312, A313, and A316, have been systematically mutated into all other 19 residues in a previous scanning mutagenesis study of the hBK channel[22]. The results reveal a striking correlation between side-chain solvation properties of the mutant with channel opening probabilities. It was thought that these residues may be involved in conformational transitions and only become exposed upon channel activation[22]. However, the Cryo-EM structures have now convincingly shown that there is minimal conformational changes in the deep-pore region between the metal-bound and metal-free states[20, 21]. Instead, the scanning mutagenesis results provide a strong support for the hydrophobic gating mechanism described above. As illustrated in Supplementary Fig. 9 for the A316 position, the half activation membrane voltage ($V_{1/2}$) measured for the wild-type and various A316 mutant hBK channels is highly correlated with the hydrophobicity of side-chain substitutions. The interpretation is that replacing A316 with more

further narrowing of the pore appears to be required for maintaining a stable dry state. As illustrated in Supplementary Fig. 7, transient dewetting transitions could occur with minimal (e.g., in c1) or limited (e.g., in c7) further narrowing of the pore structure, but these transitions do not lead to stable dewetted states. The observation that the hBK pore in the metal-free state can fully dehydrate suggests that the pore behaves more like hydrophobic instead of semihydrophobic model pores[29]. We note that F315 residues in the deep-pore region (Fig. 1), while involved in stable S6 helix packing in the metal-bound state (mainly with L312 of the neighboring subunit), become quite dynamic and can readily rotate to point the side chain into the dewetted pore (Supplementary Figs. 3c and 4), which contributes to ~5 Å reduction in the pore radius ~10 Å below the filter (Fig. 3a). Nonetheless, the dynamic nature of F315 side chains in the dry state suggests that they are not a driving force, but likely a consequence of the dewetting transition. Note that the gap between neighboring S6 helices are large enough in the metal-free state to allow penetration of hydrophobic tails of lipid molecules to fill positions previously occupied by F315 side chains

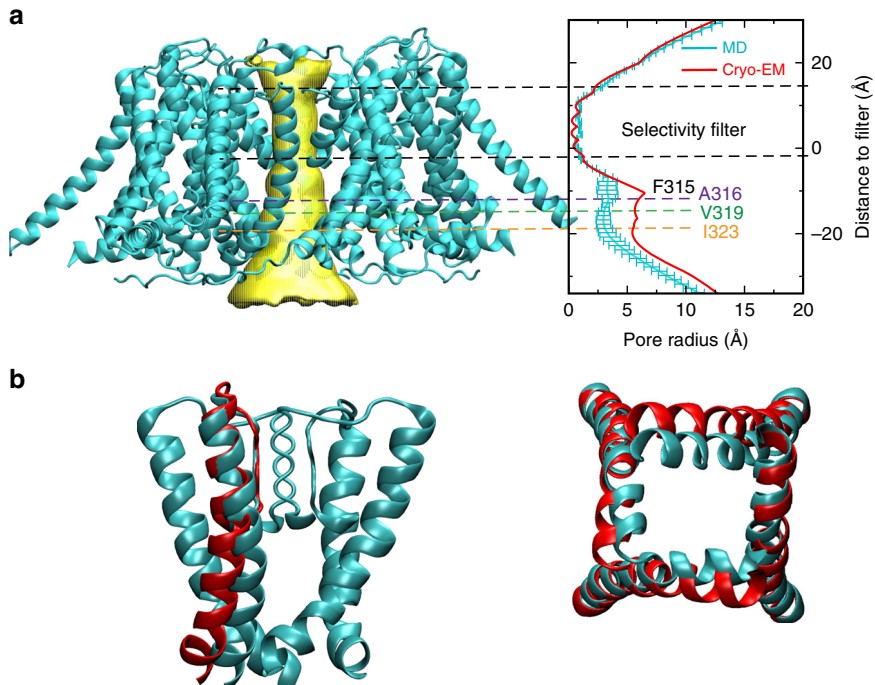

**Fig. 3** Pore profile and structural changes of the metal-free hBK channel upon dewetting transitions. **a** Representative structure of the hBK TMD in the dewetted, metal-free state (left). The pore profile calculated from HOLE27 is illustrated using the yellow funnel. The right panel compares the pore profiles derived from the initial Cryo-EM-based homology model and MD equilibrated structures. The MD profile shown is the average of snapshots from the last 50 ns of simulations $c3$, $c4$, $c6$, and $c8$, which are the most representative of the stable dry states of the pore (see Supplementary Fig. 2). The error bars reflect the standard deviations among these independent simulations. **b** Side and bottom views of S6 helices of a representative dewetted conformation of the metal-free hBK channel (cyan), in comparison with the initial homology model (red cartoon)

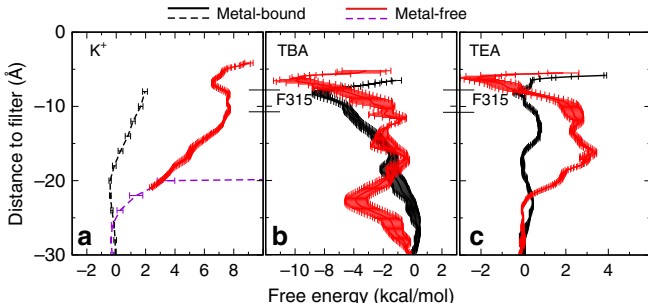

**Fig. 4** Free-energy profiles of pore permeation of $K^+$ and QAs. **a** $K^+$, **b** tetrabutylammonium (TBA), and **c** tetraethylammonium (TEA). The results calculated from umbrella sampling for the hydrated metal-bound (open) and dewetted metal-free (closed) states are shown in black and red traces, respectively. PMFs of $K^+$ derived from equilibrium simulations are shown in dashed lines in panel a (see Methods for details). The range of F315 side chain center-of-mass positions observed during equilibrium simulations is marked for reference. Error bars show standard errors between free-energy profiles from the first and second half of the data

hydrophobic residues increases the pore hydrophobicity and enhances the dewetting tendency of the pore. Conversely, polar or charged mutations reduces or completely eliminate dewetting transitions and lead to either reduced $V_{1/2}$ or constitutively open channels.

To examine if deep-pore mutants indeed modulate the pore dewetting tendency to affect the channel activation characteristics, additional atomistic simulations were performed for two contrasting hBK mutants in the metal-free state, A316D, which

is constitutively open, and A316V, which increases $V_{1/2}$ about twofold[22]. The results suggest that the A316D mutant does seem to completely prevent dewetting transitions. The pore remains fully solvated throughout all three 400-ns simulations (Fig. 5 and S10). The number of pore water molecules is consistently larger with A316D mutation, due to expansion of the pore diameter with the insertion of the charged ring. In contrast, with A316 replaced by the larger and more hydrophobic valine residue, the pore appears to undergo dewetting transitions much more readily, well within 100 ns in all the eight simulations (Supplementary Fig. 10). This is in comparison to the wild-type hBK channel, which underwent dewetting transitions within 150 ns only in four out of eight simulations (Supplementary Fig. 2) and remained semihydrated in two of the simulations for up to 800 ns. The autocorrelation function of the pore water count derived from A316V simulations decays faster and to smaller plateau values, suggesting more rapid and more complete dewetting transitions. Note that the mutant channels were similarly stable throughout the simulations, with the pore deviated only about 2 Å from the initial structures (Supplementary Fig. 11). Furthermore, free energy analysis confirms that the dewetted pore of A316V does give rise to a substantial barrier for $K^+$ permeation, while persistent hydration of A316D pore allows $K^+$ to access the selectivity filter without any substantial barrier (Supplementary Fig. 12). A strong tendency to dewet is apparently consistent with about twofold increase in $V_{1/2}$ for the A316V mutant. It is recognized that quantification of $V_{1/2}$ would require direct simulation of VSD–PGD coupling, which is much more involved computationally. Nonetheless, the above simulation and analysis provide a strong additional support for the hydrophobic gating mechanism in hBK channels.

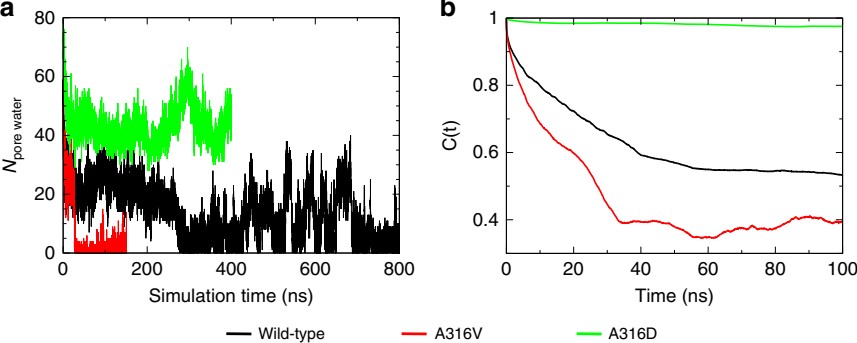

**Fig. 5** Dewetting transitions of the wild-type and mutant hBK channels. **a** The number of pore water molecules and **b** its autocorrelation function as a function of the simulation time for the wild-type, A316D and A316V hBK channels in the metal-free state

## Discussion

Extensive electrophysiological, biophysical and structural studies have so far failed to identify the physical gate in BK channels that is generally believed to be required for ion channel gating. We observe that the movement of amphipathic $V_{319}PEIIE_{324}$ segments of pore-lining S6 helices upon metal removal, as shown in recent Cryo-EM structures[15, 16], modifies key geometrical and surface properties of the pore. The resulting pore is not only narrower, with the diameter reduced from ~15 to ~10 Å, but also elongated and more hydrophobic in the metal-free state (Fig. 1d). Atomistic simulations suggest that these relatively subtle changes have important consequences on hydration of the pore central cavity. The metal-free hBK channel is able to readily undergo hydrophobic dewetting transitions to generate a dry pore that is effectively impermeable to small ions including $K^+$. Importantly, the dry pore remains physically open, with an average diameter of ~6 Å. Moderately size hydrophobic ions such as TEAs and TBAs can penetrate into the deep-pore region with minimal free energy barriers in both hydrated (open) and dry (closed) states, which provides a direct explanation for intriguing observation that QAs can block BK channels even in the closed state[15, 16]. The concept of hydrophobic gating in biological ion channels itself is not new. Hydrophobic dewetting transitions have been also shown to drive structural collapse and generate physically closed pores during deactivation of bacteria mechanosensitive channels, pentameric ligand-gated ion channels, and several tetrameric P-loop cation channels such as Kv channels[29, 30]. A key feature of hydrophobic gating as observed here for BK channels is that the dry pore in the closed state remains physically open. The barrier to $K^+$ permeation arises directly from the vapor region that separates the selectivity filter from the bulk solution. Similar gates have also been suggested to exist for several other channels[23–26].

The deep-pore region of hBK channels, lined by multiple nonpolar residues (L312, A313, F315, and A316), provide a baseline for the pore hydrophobicity, even though itself appears inadequate to trigger dewetting transitions. The gating event is triggered by initial pore reshaping, controlled independently and synergistically by metal ion binding/unbinding to CTD and transmembrane potential sensed by VSD, and involves further pore narrowing and dewetting transitions. Analysis presented in this work has emphasized the importance of the amphipathic $V_{319}PEIIE_{324}$ segment of the S6 helix in BK channel gating, particularly two key nonpolar residues V319 and I323. These two residues are involved in hydrophobic contacts to stabilize the S6 helix–helix packing in the open state (Fig. 1c). On the other hand, both residues contribute to the narrowing and elongation of the hydrophobic cavity of the pore to promote dewetting transitions in the closed state. As such, mutating V319 or I323 will affect both the open and closed states and the net consequence can be

difficult to predict. For example, V319D mutant was shown to produce no measurable current[22], presumably because of disruptive effects of charge incorporation at the V319 position to the open conformation. Mutation of I323 to Asp, which locates closer to the pore entrance, appears less disruptive to the open pore conformation, even though it compromises the stability of the open state enough to negate the reduced hydrophobicity of the pore and leads to an increase in measured $V_{1/2}$[22]. In contrast, mutation of the deep-pore hydrophobic residues, which are not involved in conformational transitions during gating, has highly predictable effects on pore dewetting tendency and channel activation properties. Taken together, the current study provides a strong support for a key role of hydrophobic gating in BK channels. Knowledge of specific protein conformational transitions involved in gating also provides a foundation for integrating computation and experiment to dissect the fundamental mechanisms of sensor-pore coupling in BK channels.

## Methods

**Homology modeling.** The structures of the hBK channel in the metal-bound and metal-free states were derived from the cryo-EM structures of the *ac*BK channel (PDB: 5tj6 and 5tji) using Modeler v9.14[32], based on the sequence alignment of Tao et al.[20]. Several long loops (C54-V91, K631-G690, and S835-T871) are absent in the Cryo-EM structures. They are presumably dynamic and not included in the current simulations (to minimize spurious nonspecific interactions with the rest of the system). The termini before and after these truncated loops are capped with either acetyl or N-methyl groups in all simulations. The initial structures of mutant hBK channels (A316D and A316V) were prepared based on the wild-type structures by replacing the original side chain with the new one in a rotamer state with minimal steric clash.

**Atomistic simulation protocols.** The homology models of the hBK channel were first inserted in POPC lipid bilayers and then solvated using TIP3P[33] water using CHARMM-GUI server[34]. The solvated systems were then neutralized by adding 450 mM KCl (as used in Cryo-EM structure determination[20, 21]). The final simulation boxes contain about ~800 lipid molecules and ~97,000 water molecules, with a total of ~476,000 atoms and dimensions of $18 \times 18 \times 15.4$ nm³. The CHARMM36m all-atom force field[35] was used for all components. The simulations were performed using CUDA-enabled versions of Amber14[36] and Gromacs 5.1.4[37, 38]. The MD time step was set at 2 fs. Electrostatic interactions were described by using the Particle Mesh Ewald algorithm[39] with a cutoff of 12 Å. Van der Waals interactions were cutoff at 12 Å with a smooth switching function starting at 10 Å. Covalent bonds to hydrogen atoms were constrained by the SHAKE algorithm[40]. The temperature was maintained at 298 K using the Nose–Hoover thermostat[41, 42] (in Gromacs) or Langevin dynamics with a friction coefficient of 1 ps$^{-1}$ (in Amber). The pressure was maintained semi-isotropically at 1 bar at both x and y (membrane lateral) directions using the Parrinello–Rahman barostat algorithm[43] (in Gromacs) or the Monte-Carlo (MC) barostat method[44, 45] (in Amber). For comparison, a separate simulation of the hBK channel was performed using the Gromos 54A7 force field[46] with the SPC water[47]. The total atom number was reduced to ~370,000 due to the united-atom representation, The system was coupled to an external temperature bath with a coupling constant of 0.1 ps using the Berendsen thermostat[48]. A reaction-field correction[49] was applied with a cutoff of 14 Å. Otherwise, equivalent setups were used as detailed above.

Each system was first minimized for 5000 steps using the steepest descent algorithm, followed by a series of equilibration steps where the positions of heavy

atoms of the protein/lipid were harmonically restrained with restrained force constants gradually decreased from 4000 to 40 kJ/(mol nm$^2$). In the last equilibration step, only protein heavy atoms were harmonically restrained and the system was equilibrated under NPT (constant particle number, pressure and temperature) conditions until the size of the simulation box became stable (which took ~5 ns for all systems in this work). Particular attention was paid to the pore region during equilibration to ensure it is fully hydrated. All production simulations were performed under NPT conditions without any positional restraints. For the wild-type hBK channel, three independent simulations of the metal-bound (open) state (o1–o3, 800 ns) and eight simulations of the metal-free (closed) state (c1–c3, 800 ns; c4–c8, 400 ns) were performed. For the A316D mutant, three simulations were performed in the metal-free state (A316D_1 to A316D_3, 400 ns), while eight simulations were performed for the metal-free state of the A316V mutant (A316V_1 to A316V_8, 150 ns). Two additional simulations of the acBK channel in both metal-bound and metal-free states were performed using the same setup for 300 ns each. The control simulation of the metal-free hBK channel in SPC water was performed for 200 ns. The cumulative length of all simulations is over 10 μs. The stability of all simulations was examined by monitoring a set of system properties including simulation box size, pressure, lipid density, and protein structure (e.g., see Supplementary Figs. 1 and 10).

**Free-energy calculations.** Umbrella sampling[50] was used to calculate the PMF of pore permeation of K$^+$, TEA, and TBA in both metal-bound (open) and/or metal-free (closed) states of hBK channels. The initial structures of open and closed hBK channel were taken from representative, fully equilibrated snapshots from unrestrained simulations (see above). A K$^+$ ion, TEA or TBA molecule was first placed in the center of the pore (~10 Å below the selectivity filter), and the initial conformations for umbrella sampling were then generated by two 5-ns steered MD simulations (upward and downward), during which the protein backbone was harmonically restrained with a force constant of 200 kJ/(mol nm$^2$). The moving harmonic pulling potential was imposed on the center-of-mass of the target molecule with a force constant of 150 kJ/(mol nm$^2$). The initial structures were carefully inspected to ensure that no substantial distortion was inadvertently introduced. Umbrella sampling windows were placed at 2 Å intervals covering 6–30 Å below the selectivity filter for TEA and TBA and 6–20 Å for K$^+$. The spring constant of 50–200 kJ/(mol nm$^2$) was used to restrain the distance along the membrane normal between the ion or small molecule and the center-of-mass of the backbone of residues 286–288 (the selectivity filter entrance). Each window was simulated for 10 ns, and the final PMFs were calculated using the weighted histogram analysis method[51]. Uncertainty of the PMF was estimated as standard errors between free-energy profiles from the first and second half of the data. We note that there is high probability of multiple K$^+$ occupancy in the hydrated metal-bound pore or ~20 Å below the selectivity filter in the metal-free state. The single-ion PMF calculated from umbrella sampling is thus only valid for the dewetted pore region from 6 to 20 Å. The overall PMFs of K$^+$ permeation for the closed state of channel from 6 to 30 Å shown in Fig. 4 were constructed from umbrella sampling and equilibrium MD using linear interpolation[52, 53].

**Analysis.** Unless stated otherwise, snapshots were extracted every 50 ps from the last 50 ns of all equilibrium MD trajectories for calculation of statistical distributions. Molecular illustrations were prepared using VMD[54]. The orientation of side chain and number of water inside the pore are analyzed using MDAnalysis[55] together with in-house scripts. Pore water molecules were identified as those occupying the internal cavity below the selectivity filter, roughly from L312 to the plane defined by the center of mass of P320 (see Fig. 3). Note that the pore is fully solvated and connected to the bulk water in the metal-bound state, while the solvated region of the dewetted pore starts around I323 in the metal-free state (Fig. 2). The kinetics of dewetting transitions is analyzed by calculating the autocorrelation function of the pore water number, $N_{water}(t)$, as $C(\tau) = \frac{<N_{water}(t)N_{water}(t+\tau)>}{<N^2_{water}>}$, where $<>$ denotes averaging overall trajectories and $\tau$ is the time lag.

**Data availability.** Data supporting the findings of this manuscript are available from the corresponding author upon reasonable request.

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

## Acknowledgments

All simulations were performed on the pikes GPU cluster housed in the Massachusetts Green High Performance Computing Cluster (MGHPCC). This work was supported by National Institutes of Health Grants R01 HL142301 (to J. Cui and J. Chen), R01 HL126774 (to J. Cui) and R01 GM114694 (to J. Cui).

## Author contributions

Conception and design of the study: Z.J., M.Y., G.Z., J. Cui, and J. Chen; Performing the simulation and analysis: Z.J. and M.Y.; Analysis and interpretation of data, drafting, and revising the manuscript: Z.J., M.Y., G.Z., J. Cui, and J. Chen.

## Additional information

**Competing interests:** The authors declare no competing interests.

