## [Peer Review File · Nature Communications]

Reviewers' comments:

Reviewer #1 (Remarks to the Author):

This is an interesting paper using state of the art molecular simulations to investigate hydrophobic gating in BK channels. As the authors state (page 12) the concept of hydrophobic gating in ion channels and model nanopores is not new. Whilst one might argue whether this is "the first bona fide hydrophobic gate reported for a biological ion channel" (page 12) the study provides clear physical insights supporting a hydrophobic gating mechanism for BK channels. This in turn provides an answer to outstanding questions as to how this physiologically important class of K channels is gated, and adds to our fundamental understanding of the relationship between channel structure, dynamics, and function. The simulations are carefully performed and analysed in some detail, and the conclusions drawn are sound. I therefore think the paper merits publication subject to a number of modifications.

1. Page 2 (abstract) & also page 4: "non-trivial changes". I am not sure what is meant by non-trivial. Do they simply mean small? I suppose a trivial change is one that does not have a functional consequence? Anyway, I would strongly suggest rewording this.
2. Page 5, figure 1 legend, last sentence. A typo: 'with 10 Å' should be 'within 10 Å'
3. Page 6, line 4: 'modest narrowing' – better to state directly how much the pore radius drops in the main text.
4. Page 6, line 16 and also the final line: 'deplete' to 'depleted'.
5. Page 6, final paragraph. It is stated that 'the pore was highly stable' for the metal bound state. Perhaps some evidence of this should be presented, e.g. an RMSD and/or pore (HOLE) radius plots vs. time for the metal bound vs. the metal free simulations to show the difference in 'stability' directly.
6. Page 7: In runs c1 and c7 the pore fluctuated and did not reach a stale dry state. Were these fluctuations just in the number of waters or also in the pore geometries? This might allow one to work out which occurs first – a change in pore geometry or dewetting.
7. Figure 2: Whilst it is ok to obtain free energy profiles for water directly from densities, I am less convinced about doing so for K⁺ ions. How well do the latter compare with the subsequent free energy profiles for ions from umbrella sampling, and are the latter not more reliable?
8. Page 8 and Figure 3: A case is made for a key role of the sidechains of F315 in narrowing and dewetting of the pore. Perhaps this sidechain should be added to Fig. 2b, right hand panel rather than being hidden away in Fig. S2c?
9. Figure 3 a. The pore radius profile averages from the MD simulations is shown – error bars should be added.
10. Figure 4: The free energy profiles are sampled every 2 Å along the pore axis. Is this fine enough sampling? I would also like to see, perhaps in the SI, some evidence of the convergence of the free energy profiles to demonstrate 10 ns per window is sufficient. Also, it would be useful to indicate the position on the z axis of key residues e.g. F315.
11. Page 9: The results for the BK channel are contrasted with those for other studies where dewetting is stated to "drive further structural collapse". Can the authors really exclude this for their channel? As they note (see comment 6 above) there is variability between simulations, so it is entirely conceivable that longer simulations (e.g. several microseconds) could result in "further structural collapse". I think they need to be less dogmatic and provide a more nuanced discussion here.
12. Page 10: Given the importance of F315 would it be useful, at least in silico, to test the effects of e.g. an F to A or an F to N mutation?

Reviewer #2 (Remarks to the Author):

General Comments: In this work sub- μ s MD simulations using two hSlo1 homologous structures

are used to address the important question of how ion permeation is gated in the BK type potassium channels. In these simulations the authors observe that water molecules may be depleted from the inner pore region of the hSlo1 structure derived from the cryo-EM Aplysia BK channel structure solved in the absence of BK channel ligands (Ca²⁺ and Mg²⁺). This "dewetted" state results in a pore that is both significantly narrower and more hydrophobic than that of the cryo-EM structure. The authors also determine that the free-energy of pore permeation for K⁺ increases substantially in the dewetted state, but not QA blockers such as TBA and TEA, potentially consistent with earlier experimental results suggesting that small QA blockers can readily access the central cavity both in open and closed conformations. Furthermore, a mutation known to enhance BK channel opening (A316D) was shown to prevent dehydration in the metal-free structure during three trials of 400 μ s simulation. Based on these results the authors propose that BK channel "does not require a physical gate", but, instead, hydrophobic gating plays a key role in BK channel gating.

This work is intriguing in that it brings into attention for the first time that hydrophobic gating may contribute to the activation of BK channel, which appears to lack a conventional "bundle-crossing" gate to provide a physical barrier of ion permeation at its cytosolic entrance. However, there are several general concerns that the authors should consider.

First, the paper may be over-interpreting the significance of the results. Specifically, the wetting/dewetting transitions happen within several tens or hundreds ns, with dry periods longer than 100-ns only observed in half of the simulations performed in the metal-free structure, while the gating of BK channel is several orders of magnitude slower. Furthermore, even for the closed conformation, channels only spend part of 500 ns in very low water states. If average water occupancy is indicative of ability of K⁺ to permeate the channel, one would really like to know the average level of water occupancy over time compared to the metal-bound channel. Visual inspection of the traces provided suggests that, based on the authors contentions, one might expect an average closed conformation current that is about 20% of the open conformation. This may be impossible to address without longer simulations, but it does probably require that some caution be taken in regards to whether dewetting fully represents closure. Ensemble averages of all 10 metal-free channel simulations might be informative. Is a steady-state plateau achieved, or will that only be reached long after 500 ns?

Second, a major change in S6 shape and position occurs during dewetting, such that the S6 in this dewetted channel no longer approximates the metal free structure. Since the simulations are done with mammalian side chains replacing their Aplysia counterparts, how are we to know that the change that occurs during dewetting arises only from the dewetting or might be a consequence of the different residues in mammalian Slo1. For example, the absence of the hinge glycine in Aplysia Slo1 may result in a very different pore architecture than occurs in human or mouse Slo1. Might any of the changes observed in these simulations reflect the artificial introduction of mammalian side chains? One test would be to run the same simulations on Aplysia Slo1. Is the same behavior observed?

Third, basing analysis and conclusions on essentially 50 ns taken from 4 out of 9 metal-free simulations is a bit like trying to take 10 ms of single channel behavior and trying to generalize about how a population of channels behaves under a variety of conditions. Although this is the state of MD at present, more caution regarding conclusions and what the ensemble behavior might be would be appreciated.

Fourth, the wording in this paper tends to overstate what conclusions can actually be drawn from simulations, e.g., citing other simulation papers as if the simulations confirm that a channel behaves in a given way. In this reader's view, a simulation can test the plausibility of a given mechanism, but it doesn't prove it. The best simulations lead to testable predictions that might distinguish between, e.g., dewetting models vs. a model involving some subtle conformational changes in the selectivity filter. Although the tests presented in this paper give support to the dewetting idea, the experiments that have been done can also be rationalized other ways. Is there

a unique prediction? Might D2O behave differently than H2O in terms of whether dehydration of the pore occurs? Are there unique temperature or pressure predictions (obviously experimentally challenging)? Making unique predictions can be difficult, but in terms of the present paper, some caution should be taken in regards to whether simulations can, at least at this point in the development of MD, be taken as reality and proof.

Specific Comments:

1. For all the simulations performed in this paper, what was the initial occupancy of K⁺ in the selectivity filter?
2. p. 3. line 3. should be "generally contains structures"
3. P. 3, ln. 11-12, "opened by depolarization of membrane potential and the binding of intracellular Ca²⁺ and Mg²⁺". Unlike Ca²⁺, Mg²⁺ enhances coupling between BK voltage-sensor and pore, but not directly increase BK activation.
4. P. 3, ln. 14, "(VSD) detects the membrane potential", perhaps should be "(VSD) detects changes in membrane potential".
5. p. 3. The discussion of QA accessibility is rather superficial. Given the rapidity of block, it can be difficult to assess for the low affinity blockers whether the channels are already in the closed channel or the blockers simply equilibrate faster than measurement time. The Wilkens and Aldrich paper nicely addressed that with a higher affinity blocker, bbTBA, but another paper (JGP 134, 409-436) that looked more closely at bbTBA block at negative potentials suggests that in a fully closed channel bbTBA may not block.
6. P. 4, ln. 24, "the first example of a true hydrophobic gate". There is a 2014 Nature communication paper about hydrophobic gating in the TWIK-1 K2P channel that adopts an open conformation at its cytosolic entrance (Aryal P. etc, A hydrophobic barrier deep within the inner pore of the TWIK-1 K2P potassium channel)
7. p. 6. "Hydrophobic dewetting has been shown to underlie". This is an example of what might be considered an overstatement. To my knowledge, there is no actual experimental data showing that hydrophobic dewetting actually occurs in a channel.
8. p. 12. paragraph at top is another example where statements are made that imply there is experimental evidence that hydrophobic dewetting transitions can do certain things. What the simulations show that, in principle, structural collapse of a pore may occur as a consequence of dewetting transitions.
9. p. 12. "because of disruptive effects of charges incorporation" should be "charge"
10. p. 13. NPT is never defined.
11. p. 13. near bottom, "was equilibrated in under NPT conditions" omit "in"
12. Figure 1 C, an additional view from cytosolic entrance may help to visualize the difference between the two structures as described in the text (p. 5, para. 1).
13. Figure 3A shows a remarkable conformational change in the dry pore. What about partially dehydrated pore? In addition, is there any noticeable change in other part of the channel, such as the link between pore and gating ring? If one runs the simulation with the Aplysia side chains, does this happen?
14. Steered MD (SMD) was used to generate initial conformation for free-energy calculation of TEA/TBA permeation shown in Figure 4. Assuming it was constant velocity SMD, plots of force vs. position should be provided to show that no extra-large force was required to pull TEA or TBA into each sampling window.
15. Figure 5. Is it possible to also include free-energy profile of ion permeation for A316V and A316D?
16. What method was used to induce these mutations?
17. P. 13, para. 1, since there are three long loops missing from the homologous structures, each subunit in these structures is formed by four segments that are not covalently connected. Is there any constrain applied to these segments to keep them in position during simulation?
18. P. 14, para. 2, which atom(s) in TEA or TBA was pulled during SMD? In addition, the permeation molecule was placed 10Å below the selectivity filter before each simulation, but

umbrella sampling windows cover from 6 to 30 Å. Does this mean that two SMDs toward opposite directions were performed for each umbrella sampling?

19. P. 15, ln. 2, more details and/or proper reference should be provided for the auto-correlation analysis. Also, in the "a C(τ)" equation it isn't clear what "a" is doing here.

20. Figure S7. The metal free simulations A316V_1 and A316V_2 appear impossibly identical. Is there a problem with any random number generation used in these simulations or was the same simulation displayed twice?

21. Figure S3. The legend in the plot (Metal-bound (run:o1), metal-free (run:c3)) indicate that the orientation of F315 was calculate from one run of simulation. Why there are several lines in different colors in each plot? In addition, the orientation of F315 is not specifically defined. It is only mentioned in the legend of Figure S2 that "Smaller angles correspond to the cases of side chain pointing toward the center of the pore."

Reviewer #3 (Remarks to the Author):

This paper by Jia et al provides convincing supporting evidence for a hydrophobic gating mechanism in the BK K⁺ channels. The computational approaches used are generally well conducted and convincing. Overall this study has the potential to make an important contribution to our understanding of the structural and biophysical mechanisms which underlie gating of these K⁺ channels. However, before I am able to fully support the conclusions drawn from this study there are a number of issues that the authors need to address and/or answer.

Major

1. The authors keep referring to various effects seen upon 'metal removal'. As far as I can see, this term is not as it suggests and is potentially misleading. The authors appear to be only showing that there are differences between simulations of the 'metal-bound' and 'metal-free' states i.e. they do not show a dynamic change in behaviour of one single state (or simulation) where the metal has been removed (or added) after a certain period of time. This may seem a trivial point, but precise descriptions matter. In fact, the study might be much more convincing if the authors took the stable, wet simulations and removed the metal to see if there is a subsequent change – has this been done?

2. The authors make bold claims about this study being the first example of a 'true' hydrophobic gate. It is not. There are other cases where rapid pore dewetting of the nanocavity occurs without structural collapse of the pore including in several LGICS (e.g. nAChR, Beckstein PNAS 2006; 5HTR, Trick et al Structure 2016) and the TWIK K⁺ channel (Aryal et al Nat Comms 2014) – in all these cases the pore remains open, but becomes dewetted. In fact the hydrophobic-induced collapse they refer to in Kv channels seems to be the exception. In addition they should also refer to the recent work on hydrophobic gating mechanisms in TRP channels (Zheng FASEB 2018) and CRAC channels (Yamashita, Nat Comms 2017).

This does not detract from the importance of the authors novel findings for the BK channel - it even helps support their case as well as explaining so much previous data from the BK field. But they should acknowledge these previous findings and cite them appropriately.

3. No data is provided for the quality of the homology models that are made. How stable are they – no RMSD/RMSF information is provided – they refer to several long loops being missing in their model - with the regions simply being capped – yet unrestrained? How stable are their models during the simulations?

4. The dewetting also only occurs as a consequence of the structural changes that occur in their homology model of the metal-free state. The authors infer that it is due to 'metal removal' but this is not directly shown. I assume if the structures are restrained then no dewetting occurs in either state? Could it simply be due to the fact that the homology model of the metal free state is less

stable?

Minor points

5. Fig 1d the labels are not clear at all.

6. p7. Top. Have the authors attempted to precisely correlate the structural movements with dewetting transitions i.e.. Are there any situations in which S6 moves but dewetting does not occur?

7. Fig 2. It does not seem to be stated clearly which area is actually measured to calculate Nporewater – it might be helpful to show as a box on panel b?

8. p8 , the authors mention the role that F315 plays – it seems to be responsible for a large change in diameter – does functional data exist for mutants at this position?

9. Fig3. Needs to be clearer that the MD equilibrated structures shown are of the metal free state (I assume?)

10. p9 ...'represent true gating events'. The authors need to be careful in their choice of language – they observe dewetted states that are very likely to be non-conductive, 'true' is probably too strong a statement. Likewise, the next sentence is also incorrect – open dewetted pores have been seen before in other channels (see above).

11. p10 In the discussion about mutations - in MthK, mutations in a similar region produce a change in single channel conductance, is this also the case for the A316 mutns. Could this provide any insight into the mechanisms involved?

12. How stable is the A316D mutation – I assume no structural collapse occurs here but that it does in A316V – is there a correlation with movements and dewetting for these mutations – no RMSDs are shown (even for just this region). Introducing an Asp into this position may also cause repulsion and prevent collapse – did they try any other polar mutns at this position e.g. A316T or A316N?

13. p12. This is not the first description of a bona fide hydrophobic gate (see above).

14. methods – the choice of TIP3P water is appropriate for the FF used – but did they try any other water types?

Responses to Reviewers' Comments (all original comments shown in *Italic*; Corresponding changes to the revised manuscript are also highlighted using Track Change function of Word)

Reviewer #1 (Remarks to the Author):

This is an interesting paper using state of the art molecular simulations to investigate hydrophobic gating in BK channels. As the authors state (page 12) the concept of hydrophobic gating in ion channels and model nanopores is not new. Whilst one might argue whether this is "the first bona fide hydrophobic gate reported for a biological ion channel" (page 12) the study provides clear physical insights supporting a hydrophobic gating mechanism for BK channels. This in turn provides an answer to outstanding questions as to how this physiologically important class of K channels is gated, and adds to our fundamental understanding of the relationship between channel structure, dynamics, and function. The simulations are carefully performed and analysed in some detail, and the conclusions drawn are sound. I therefore think the paper merits publication subject to a number of modifications.

Responses: We greatly appreciate the reviewer's positive and insightful comments.

1. Page 2 (abstract) & also page 4: "non-trivial changes". I am not sure what is meant by non-trivial. Do they simply mean small? I suppose a trivial change is one that does not have a functional consequence? Anyway, I would strongly suggest rewording this.

Responses: The phrase "non-trivial" has been deleted in the abstract.

2. Page 5, figure 1 legend, last sentence. A typo: 'with 10 Å' should be 'within 10 Å'

Responses: This has been corrected as suggested.

3. Page 6, line 4: 'modest narrowing' – better to state directly how much the pore radius drops in the main text.

Responses: The "modest narrowing" of the pore has been discussed in detailed in the published Cryo-EM structural paper (MacKinnon and cow-workers, 2017); it is not uniform throughout the pore region (a portion of the "deep pore" actually become wider in the metal-free structure; see Fig. 1b). Therefore, it is potentially misleading to quantify how much the pore radius drops. Instead, we retain the original "modest narrowing" wording and add a note "by -1 to 4 Å at various locations as previously noted" with references at the end of the sentence for clarification (see **Page 6 line 4-5**).

4. Page 6, line 16 and also the final line: 'deplete' to 'depleted'.

Responses: The error has been corrected in the location noted above as well as another occurrence of "deplete".

5. Page 6, final paragraph. It is stated that 'the pore was highly stable' for the metal bound

state. Perhaps some evidence of this should be presented, e.g. an RMSD and/or pore (HOLE) radius plots vs. time for the metal bound vs. the metal free simulations to show the difference in 'stability' directly.

Responses: A new figure (**Fig S1**) has been added to SI. The new Fig S1 shows that the whole channel remains below 0.5 nm RMSD from the starting structures throughout the simulations. The RMSD for the pore forming S6 helices is less than ~0.2 nm. The main text on **Page 6 (final paragraph)** has been revised accordingly to include a discussion of the new figure.

6. Page 7: In runs c1 and c7 the pore fluctuated and did not reach a stable dry state. Were these fluctuations just in the number of waters or also in the pore geometries? This might allow one to work out which occurs first – a change in pore geometry or dewetting.

Responses: We thank the reviewer for the excellent suggestion. We have performed additional analysis to examine the structural origin of different dewetting behaviors in these simulations. The results, summarized in a new **Fig. S7**, suggest that formation of stable dewetted states require further structural narrowing as discussed in **Page 8-9 (texts between Figs 2 and 3)** and illustrated in **Fig. 3b**. Transient dewetting transitions could occur with limited further narrowing of the pore structure, but these transitions do not appear to lead to a stable dewetted state (**Fig. S4**). The main text has also been modified to include a similar discussion of the new analysis (**bottom of Page 8 to the beginning of Page 9**).

7. Figure 2: Whilst it is ok to obtain free energy profiles for water directly from densities, I am less convinced about doing so for K⁺ ions. How well do the latter compare with the subsequent free energy profiles for ions from umbrella sampling, and are the latter not more reliable?

Responses: We agree that densities from equilibrium simulations will not be sufficient for estimating the free energy profile of K⁺ ion permeation. The profiles shown in Fig. 2c, derived directly from equilibrium distributions, are mainly to illustrate that K⁺ ions are excluded from the pore together with water upon dewetting transition. The complete free energy profiles are shown in Fig. 4a, which require both equilibrium simulations (for multi-ion regions outside the pore) and umbrella sampling (for single-ion region inside the pore). Details of the free energy calculation protocol are provided in **Methods** under section “**Free energy calculations**” (**Pages 15-16**).

8. Page 8 and Figure 3: A case is made for a key role of the sidechains of F315 in narrowing and dewetting of the pore. Perhaps this sidechain should be added to Fig. 2b, right hand panel rather than being hidden away in Fig. S2c?

Responses: We had actually also considered including Fig. S2c (now Fig. S3c) in the main text as the reviewer suggested. However, detailed analysis suggests that the movement of F315 side chains, even though responsible for substantial narrowing of the deep pore region, does not appear consequential in the dewetting (and deactivation) transition. Instead, dynamics of the F315 side chains appears more likely a consequence instead of a driving

force of dewetting. A similar statement has been added to the main text where dynamics of F315 is discussed (**Page 9, lines 6-8**).

9. *Figure 3 a. The pore radius profile averages from the MD simulations is shown – error bars should be added.*

Responses: Error bars have been added to MD-derived pore profiles in both **Fig 3a and S4**.

10. *Figure 4: The free energy profiles are sampled every 2 Å along the pore axis. Is this fine enough sampling? I would also like to see, perhaps in the SI, some evidence of the convergence of the free energy profiles to demonstrate 10 ns per window is sufficient. Also, it would be useful to indicate the position on the z axis of key residues e.g. F315.*

Responses: Analysis of convergence, as shown in the error bars in **Fig. 4**, suggest that the umbrella sampling protocol is adequate to derive well-converged free energy profiles. The approximate location of F315 has been added to **Panels b & c**.

11. *Page 9: The results for the BK channel are contrasted with those for other studies where dewetting is stated to "drive further structural collapse". Can the authors really exclude this for their channel? As they note (see comment 6 above) there is variability between simulations, so it is entirely conceivable that longer simulations (e.g. several microseconds) could result in "further structural collapse". I think they need to be less dogmatic and provide a more nuanced discussion here.*

Responses: We have toned down the discussion throughout to the text to reflect a cautious interpretation of the simulation trajectories (also see responses to Reviewer 2's comments below). The notion of a physically open deactivated state of the pore revealed by the current computational studies appears to reconcile a set of existing experimental data, including the channel blocker experiments (**Fig. 4 b-c; page 10-11**), pore scanning mutagenesis studies (**Figs. 5, S8-9; pages 11-12**) and cysteine scanning and modification data (**Ref. 17**). The comparison between simulation and experimental data suggests that the pore in BK channels is unlikely to undergo further collapse even with much longer simulations (assuming that there is no major artifact associated with the force fields).

12. *Page 10: Given the importance of F315 would it be useful, at least in silico, to test the effects of e.g. an F to A or an F to N mutation?*

Responses: While we agree that F315 is very interesting, it does not appear to play a driving role in the dewetting transition (see response to comment #8 above). Nonetheless, several mutants (e.g., F315A, F315W) have been previously shown to have important consequence on gating of BK channels (Carrasquel-Ursulaez et al, Journal of General Physiology, 2015). Currently, there is no experimental data for F315N. Some of F315 mutants are being investigated in our labs and the results will be reported in a separate article.

Reviewer #2 (Remarks to the Author):

*General Comments: In this work sub- μ s MD simulations using two hSlo1 homologous structures are used to address the important question of how ion permeation is gated in the BK type potassium channels. In these simulations the authors observe that water molecules may be depleted from the inner pore region of the hSlo1 structure derived from the cryo-EM *Aplysia* BK channel structure solved in the absence of BK channel ligands (Ca^{2+} and Mg^{2+}). This "dewetted" state results in a pore that is both significantly narrower and more hydrophobic than that of the cryo-EM structure. The authors also determine that the free-energy of pore permeation for K^+ increases substantially in the dewetted state, but not QA blockers such as TBA and TEA, potentially consistent with earlier experimental results suggesting that small QA blockers can readily access the central cavity both in open and closed conformations. Furthermore, a mutation known to enhance BK channel opening (A316D) was shown to prevent dehydration in the metal-free structure during three trials of 400 μ s simulation. Based on these results the authors propose that BK channel "does not require a physical gate", but, instead, hydrophobic gating plays a key role in BK channel gating.*

This work is intriguing in that it brings into attention for the first time that hydrophobic gating may contribute to the activation of BK channel, which appears to lack a conventional "bundle-crossing" gate to provide a physical barrier of ion permeation at its cytosolic entrance. However, there are several general concerns that the authors should consider.

Responses: We are very grateful to the reviewer for insightful comments and constructive critiques!

First, the paper may be over-interpreting the significance of the results. Specifically, the wetting/dewetting transitions happen within several tens or hundreds ns, with dry periods longer than 100-ns only observed in half of the simulations performed in the metal-free structure, while the gating of BK channel is several orders of magnitude slower. Furthermore, even for the closed conformation, channels only spend part of 500 ns in very low water states. If average water occupancy is indicative of ability of K to permeate the channel, one would really like to know the average level of water occupancy over time compared to the metal-bound channel. Visual inspection of the traces provided suggests that, based on the authors contentions, one might expect an average closed conformation current that is about 20% of the open conformation. This may be impossible to address without longer simulations, but it does probably require that some caution be taken in regards to whether dewetting fully represents closure. Ensemble averages of all 10 metal-free channel simulations might be informative. Is a steady-state plateau achieved, or will that only be reached long after 500 ns?

Responses: We appreciate the note that gating of BK channel likely occurs at timescales several orders of magnitude slower than those of dewetting transitions observed. We expect that channel deactivation starts with conformational changes in the pore (as induced by removing the metal from the RCK domains of the gating ring) and then the narrower pore

triggers dewetting and eventual blockage of K⁺ permeation. All simulations of BK channels were initiated from Cryo-EM-derived homology models, such that the major conformational changes of the pore induced by metal unbinding, which are presumably slow, are not included in the simulation directly (please also see the response to **Reviewer 3's comment #1**). In other words, the simulation skipped the rate limiting step of metal-unbinding induced conformational changes during deactivation, but was primarily on the dewetting step. This is why the simulation results are much faster than the deactivation kinetics of the channel.

For the second point of the reviewer's comment above, we note that all simulations of BK channels were initiated from fully solvated, Cryo-EM-derived homology models. Dewetting transitions from such initial states appear to require hundreds of ns to multi μ s in order to reach a stable dry state (discussed in **pages 8-9**; also see the response to **Reviewer 1 point #6** above). As such, runs of 400 – 800 ns in lengths here are not expected to provide a reliable estimate of the life time or occupancy of the dry state. We are planning to perform free energy calculations using biased simulations in order to provide a more reliable estimate of the open/close transition thermodynamics. These studies are beyond the scope of the current work and will be reported in a forthcoming publication.

We have also more carefully analyzed the molecular origin of the observed slow transitions, particularly, why in some simulations dewetting did not occur with up to 800 ns (**Fig. S2, runs c2 and c5**) or failed to generate a stable dry pore (**Fig. S2, runs c1 and c7**). The analysis showed that that further structural narrowing, as detailed in **Figs. 3 and S3**, is required for reaching a stable dry state. The new analysis is summarized in a **new Fig. S7**, and discussed on **Page 8, last two lines to Page 8 first two lines**.

Second, a major change in S6 shape and position occurs during dewetting, such that the S6 in this dewetted channel no longer approximates the metal free structure. Since the simulations are done with mammalian side chains replacing their Aplysia counterparts, how are we to know that the change that occurs during dewetting arises only from the dewetting or might be a consequence of the different residues in mammalian Slo1. For example, the absence of the hinge glycine in Aplysia Slo1 may result in a very different pore architecture than occurs in human or mouse Slo1. Might any of the changes observed in these simulations reflect the artificial introduction of mammalian side chains? One test would be to run the same simulations on Aplysia Slo1. Is the same behavior observed?

Responses: This is a great suggestion. We have now performed additional control simulations of Aplysia Slo1 in both metal-bound and metal-free states using identical protocols. The results are summarized in a **new Fig. S6a**; they confirm that Aplysia Slo1 also readily undergoes dewetting transition in the metal free state. Therefore, the ability to undergo dewetting transition appears to be conserved among different BK channels. Discussion of these new simulations and results have been included in the revised manuscript (**page 7, lines 14-16, and Page 14, lines 17-19**).

Third, basing analysis and conclusions on essentially 50 ns taken from 4 out of 9 metal-free simulations is a bit like trying to take 10 ms of single channel behavior and trying to

generalize about how a population of channels behaves under a variety of conditions. Although this is the state of MD at present, more caution regarding conclusions and what the ensemble behavior might be would be appreciated.

Responses: As discussed in the response to comment #1 above, all simulations were initiated from fully hydrated states. They do not all reached stable dry states within 400-800 ns, due to the apparently slow conformational transitions required for maintaining a stable dry pore. Therefore, the last 50 ns segments of runs c3, c4, c6 and c8 are likely the most representative of the predicted dry, deactivated state of the pore. **The caption of Fig. 3** has been slightly revised to reflect this rationale (**line 5 of the caption**).

Fourth, the wording in this paper tends to overstate what conclusions can actually be drawn from simulations, e.g., citing other simulation papers as if the simulations confirm that a channel behaves in a given way. In this reader's view, a simulation can test the plausibility of a given mechanism, but it doesn't prove it. The best simulations lead to testable predictions that might distinguish between, e.g., dewetting models vs. a model involving some subtle conformational changes in the selectivity filter. Although the tests presented in this paper give support to the dewetting idea, the experiments that have been done can also be rationalized other ways. Is there a unique prediction? Might D2O behave differently than H2O in terms of whether dehydration of the pore occurs? Are there unique temperature or pressure predictions (obviously experimentally challenging)? Making unique predictions can be difficult, but in terms of the present paper, some caution should be taken in regards to whether simulations can, at least at this point in the development of MD, be taken as reality and proof.

Responses: We fully agree with the notions that biomolecular simulations alone generally could not establish a biological mechanism (particularly with the current limitations in both simulation timescales and force field accuracy) and that one of the most valuable outputs of simulation is new insights and hypotheses that can guide further computational and/or experimental investigation. We have carefully revised the discussion throughout the main text to more precisely reflect the computational nature of this work and that additional work is needed to further establish the proposed hydrophobic gating mechanism in BK channels.

Specifically for this work, the novel hypothesis derived from atomistic simulations is that gating of BK channels may not require a physically closed barrier as previously thought. Even though new experiments have not been described as part of this work, the new hypothesis has allowed us to re-examine the interpretation of several exiting experimental studies, including channel blocker experiments, pore scanning mutagenesis studies and cysteine scanning and modification data. Additional calculations were performed to verify that: 1) the dry pore indeed gives rise to K⁺ permeation barrier whiling allowing access by hydrophobic ions (see **Fig. 4; page 10-11**). 2) Effects of pore mutations on channel activation can be correlated with the tendency of pore dewetting (see **Figs. 5, S8-9; pages 11-12**). The hydrophobic gating hypothesis proposed here may also lead to other predictions that are testable by experiment and computation, which will be performed in follow-up works. We note that isotope effects are expected to be minimal in hydrophobic dewetting transitions.

Specific Comments:

1. For all the simulations performed in this paper, what was the initial occupancy of K⁺ in the selectivity filter?

Responses: The selective filter is fully occupied in all initial structures, with K⁺ ions at S0-S4 sites. K⁺ ions in S1-S3 remain stably bound throughout all simulations, but undergo dynamic exchanges at S0 and S4 sites. The filter itself remains extremely stable in all simulations, with backbone RMSD fluctuating stably around ~0.5 Å (see **new Fig. S1c**). Additional discussion of the filter is now included in **Page. 7 lines 11-15**.

2. p. 3. line 3. should be "generally contains structures"

Responses: Thank you very much for thorough reading. It has been corrected as suggested.

3. P. 3, ln. 11-12, "opened by depolarization of membrane potential and the binding of intracellular Ca²⁺ and Mg²⁺". Unlike Ca²⁺, Mg²⁺ enhances coupling between BK voltage-sensor and pore, but not directly increase BK activation.

Responses: The above sentence has been rephrased as suggested to more precisely describe the roles of intracellular Mg²⁺.

4. P. 3, ln. 14, "(VSD) detects the membrane potential", perhaps should be "(VSD) detects changes in membrane potential".

Responses: This has been corrected as suggested.

5. p. 3. The discussion of QA accessibility is rather superficial. Given the rapidity of block, it can be difficult to assess for the low affinity blockers whether the channels are already in the closed channel or the blockers simply equilibrate faster than measurement time. The Wilkens and Aldrich paper nicely addressed that with a higher affinity blocker, bbTBA, but another paper (JGP 134, 409-436) that looked more closely at bbTBA block at negative potentials suggests that in a fully closed channel bbTBA may not block.

Responses: We thank the reviewer for bringing our attention to bbTBA. This is part of our ongoing follow-up work to further test the hydrophobic gating hypothesis. Preliminary free energy profiles of bbTBA permeation, shown above, suggest that bbTBA too large to enter the dry BK pore in the metal-free state. This is apparently consistent with the work by Lingle et al showing that partial activation is required for blocking BK channels by bbTBA.

6. P. 4, ln. 24, "the first example of a true hydrophobic gate". There is a 2014 Nature

Fig. Free energy profiles of bbTBA through the pore of metal-bound and metal-free (dry) hBK channel calculated using umbrella sampling.

communication paper about hydrophobic gating in the TWIK-1 K2P channel that adopts an open conformation at its cytosolic entrance (Aryal P. etc, A hydrophobic barrier deep within the inner pore of the TWIK-1 K2P potassium channel)

Responses: We appreciate the reviewer bringing this important work to our attention. We have now added the above reference and others noted by Reviewer 3 (**new Ref. 23-26**). Related discussions have also been updated to properly reflect that fact that similar hydrophobic gates have been previously observed (see **Page 4, near the bottom; Page 6, lines 19; Page 10, end of first paragraph; Page 13, end of first paragraph, etc.**).

7. p. 6. "Hydrophobic dewetting has been shown to underlie". This is an example of what might be considered an overstatement. To my knowledge, there is no actual experimental data showing that hydrophobic dewetting actually occurs in a channel.

Responses: The above phrase has been revised by replacing “shown” with “suggested”, and expanded to include a more detailed discussion of previously computational work suggesting a role of hydrophobic dewetting in multiple channels (see **Page 6, lines 18-19**).

8. p. 12. paragraph at top is another example where statements are made that imply there is experimental evidence that hydrophobic dewetting transitions can do certain things. What the simulations show that, in principle, structural collapse of a pore may occur as a consequence of dewetting transitions.

Responses: We have rephrased these statements, such as replacing “reveal” with “suggest”, inserting “seems to be able to” before “undergo”, to more clearly reflect that these are predictions and not established facts (see **the top of Page 13**).

9. p. 12. "because of disruptive effects of charges incorporation" should be "charge"

Responses: This has been corrected as suggested.

10. p. 13. NPT is never defined.

Responses: NPT (constant particle number, pressure and temperature) is now defined.

11. p. 13. near bottom, "was equilibrated in under NPT conditions" omit "in"

Responses: This has been corrected as suggested.

12. Figure 1 C, an additional view from cytosolic entrance may help to visualize the difference between the two structures as described in the text (p. 5, para. 1).

Responses: We appreciate the suggestion. In fact, it proves challenging to find a perfect perspective to illustrate the structural re-arrangements of S6 helices and how the pore geometry is affected. Fig. 1c and 1d are two of the best ways that we have identified. The view from the cytosolic entrance is actually not as helpful in visualizing these changes.

13. Figure 3A shows a remarkable conformational change in the dry pore. What about partially dehydrated pore? In addition, is there any noticeable change in other part of the channel, such as the link between pore and gating ring? If one runs the simulation with the *Aplysia* side chains, does this happen?

Responses: As detailed above in **the response to Reviewer 1's point #6**, the partially dehydrated and transient dry states apparently involve pores that do not further narrow as detailed in **Fig. 3**. The rest of the channel appears to be very stable (see the **new Fig. S1**). Specifically, there is no remarkable change in the linker region; more detailed analysis of the C-linker (particularly related to the “passive spring” model) is on-going and will be reported in a separate forthcoming manuscript.

14. Steered MD (SMD) was used to generate initial conformation for free-energy calculation of TEA/TBA permeation shown in Figure 4. Assuming it was constant velocity SMD, plots of force vs. position should be provided to show that no extra-large force was required to pull TEA or TBA into each sampling window.

Responses: The pulling force vs position from a single pulling simulation of 5 ns duration is inherently noisy showing very large fluctuations in the instantaneous force. We note that only a very weak moving harmonic potential with a force constant of 150 KJ/mol/nm² was required to pull TEA/TBA out of the deep pore during SMD. It is unlikely that extra-large force was required during pulling. We have also carefully inspected all initial structures to make sure that no substantial distortion was inadvertently introduced. These additional details are now included in the revised manuscript (**page 15, last paragraph**).

15. Figure 5. Is it possible to also include free-energy profile of ion permeation for A316V and A316D?

Responses: We have performed new umbrella sampling simulations to calculate the free energy profile of potassium permeation through the A316V mutant BK channel in the dewetted, closed state. For A316D, the pore remains hydrated and accessible to K⁺ even in the metal-free state and the free energy profile was calculated directly from equilibrium distributions. The results are summarized in **a new figure (Fig. S12)** to avoid over-crowding Fig. 5. Additional discussion of these new results is also included in **Page 12, lines 8-10**.

16. What method was used to induce these mutations?

Responses: A statement has been added in **Methods/Homology modeling** to clarify how the initial structures of mutant hBK channels were constructed (see **Page 14**)

17. P. 13, para. 1, since there are three long loops missing from the homologous structures, each subunit in these structures is formed by four segments that are not covalently connected. Is there any constrain applied to these segments to keep them in position during simulation?

Responses: The missing loops are likely dynamic and do not seem to contribute to the structural integrity of the whole channel. The remain segments have extensive structural contacts and no additional restrain was needed. As shown in **the new Fig. S1**, the channel remained highly stable in all simulations.

18. P. 14, para. 2, which atom(s) in TEA or TBA was pulled during SMD? In addition, the permeation molecule was placed 10Å below the selectivity filter before each simulation, but umbrella sampling windows cover from 6 to 30 Å. Does this mean that two SMDs toward opposite directions were performed for each umbrella sampling?

Responses: Yes, two SMD simulations were performed to pull the target molecule either towards or away from the filter. The pulling potential was imposed on the center of mass. The description for the free energy simulation protocol has been updated to include these additional details (see **Methods/Free energy calculations, pages 15-16**)

19. P. 15, ln. 2, more details and/or proper reference should be provided for the auto-correlation analysis. Also, in the "a C(τ)" equation it isn't clear what "a" is doing here.

Responses: The definition for the autocorrelation function is a standard one in statistical mechanics that should not require an additional reference. "a" is a typo; it should be "as". We have revised the description to clarify that τ is the time lag (see **page 16**, end of section "Analysis").

20. Figure S7. The metal free simulations A316V_1 and A316V_2 appear impossibly identical. Is there a problem with any random number generation used in these simulations or was the same simulation displayed twice?

Responses: We thanks the reviewer for catching the mistake. The first simulation was inadvertently presented twice during figure preparation. This has been corrected (see **revised Fig. S10**).

21. Figure S3. The legend in the plot (Metal-bound (run:o1), metal-free (run:c3)) indicate that the orientation of F315 was calculate from one run of simulation. Why there are several lines in different colors in each plot? In addition, the orientation of F315 is not specifically defined. It is only mentioned in the legend of Figure S2 that "Smaller angles correspond to the cases of side chain pointing toward the center of the pore."

Responses: The caption of this figure (**now Fig. S4**) has been revised to clarify that each trace corresponds to one subunit of the tetramer from the same simulation and include the definition of F315 side chain orientation of F315 sidechain as shown in this figure.

Reviewer #3 (Remarks to the Author):

This paper by Jia et al provides convincing supporting evidence for a hydrophobic gating mechanism in the BK K⁺ channels. The computational approaches used are generally well conducted and convincing. Overall this study has the potential to make an important contribution to our understanding of the structural and biophysical mechanisms which underlie gating of these K⁺ channels. However, before I am able to fully support the conclusions drawn from this study there are a number of issues that the authors need to address and/or answer.

Responses: We thank the reviewer for the positive and insightful comments.

Major

1. The authors keep referring to various effects seen upon 'metal removal'. As far as I can see, this term is not as it suggests and is potentially misleading. The authors appear to be only showing that there are differences between simulations of the 'metal-bound' and 'metal-free' states i.e. they do not show a dynamic change in behaviour of one single state (or simulation) where the metal has been removed (or added) after a certain period of time. This may seem a trivial point, but precise descriptions matter. In fact, the study might be much more convincing if the authors took the stable, wet simulations and removed the metal to see if there is a subsequent change – has this been done?

Responses: We fully agree with the comments. We have revised the manuscript to remove all reference to “metal removal” except when discussing the CryoEM structures. We have not attempted to directly simulate the conformational transitions induced by metal removal. There are significant conformational changes induced by metal removal throughout the channel and it is highly unlikely that atomistic simulations could recapitulate these changes due to sampling (and force field accuracy) limitations.

2. The authors make bold claims about this study being the first example of a 'true' hydrophobic gate. It is not. There are other cases where rapid pore dewetting of the nanocavity occurs without structural collapse of the pore including in several LGICS (e.g. nAchR, Beckstein PNAS 2006; 5HTR, Trick et al Structure 2016) and the TWIK K⁺ channel (Aryal et al Nat Comms 2014) – in all these cases the pore remains open, but becomes dewetted. In fact the hydrophobic-induced collapse they refer to in Kv channels seems to be the exception. In addition they should also refer to the recent work on hydrophobic gating mechanisms in TRP channels (Zheng FASEB 2018) and CRAC channels (Yamashita, Nat Comms 2017).

This does not detract from the importance of the authors novel findings for the BK channel - it even helps support their case as well as explaining so much previous data from the BK field. But they should acknowledge these previous findings and cite them appropriately.

Responses: We sincerely apologize for being unaware of these important prior works that also suggested hydrophobic dewetting without complete structural collapse. Indeed, these studies help to make a strong case that BK channels likely do not require a physically closed

pore for gating. All references mentioned above have been added to the revised manuscript (**new Ref. 23-26**), and appropriate discussions have been added (see **Page 4, near the bottom; Page 6, lines 19; Page 10, end of first paragraph; Page 13, end of first paragraph, etc.**).

3. No data is provided for the quality of the homology models that are made. How stable are they – no RMSD/RMSF information is provided – they refer to several long loops being missing in their model - with the regions simply being capped – yet unrestrained? How stable are their models during the simulations?

Responses: The termini before and after the truncated loops are capped with either acetyl or N-methyl groups in all simulations (noted in **Methods/Homology Modeling, page 14**). The evolution of backbone RMSD values from the initial structures for the whole channel, pore-forming S6 helices, and the selectivity filter are now provided in a **new Fig. S1**. The main text on **Page 6 (final paragraph)** has been revised accordingly to include a discussion of the new figure.

4. The dewetting also only occurs as a consequence of the structural changes that occur in their homology model of the metal-free state. The authors infer that it is due to 'metal removal' but this is not directly shown. I assume if the structures are restrained then no dewetting occurs in either state? Could it simply be due to the fact that the homology model of the metal free state is less stable?

Responses: Please see response to **Major Comment #1** above about reference of “metal removal” in the manuscript. All production simulations of the metal-bound and metal-free BK channels were performed without any restraints (see **Methods, section “Atomistic simulation protocols”, pages 14-15**).

Please see response to **Reviewer 2 comment #2** above for additional control simulations of Aplysia Slo1 in both metal-bound and metal-free states. The results, summarized in a **new Fig. S6a**, show that the observed dewetting transitions are not likely due to artifacts introduced by homology modeling.

Minor points

5. Fig 1d the labels are not clear at all.

Responses: The labels have been replaced with **bold fonts in purple** to improve clarity.

6. p7. Top. Have the authors attempted to precisely correlate the structural movements with dewetting transitions i.e.. Are there any situations in which S6 moves but dewetting does not occur?

Responses: As detailed above in **the response to Reviewer 1’s point #6**, we have performed additional analysis to examine the structural origin of different dewetting behaviors in these simulations. The results, summarized in a **new Fig. S7**, suggest that formation of stable

dewetted states require further structural narrowing; dewetting transitions could occur with minimal or limited further narrowing of the pore structure, but they do not appear to lead to a stable dewetted state (**Fig. S4**). We did not observe a situation where S6 moves (substantially inward) without dewetting transitions.

7. *Fig 2. It does not seem to be stated clearly which area is actually measured to calculate $N_{\text{porewater}}$ – it might be helpful to show as a box on panel b?*

Responses: The pore region used for counting the number of pore water molecules is defined in **Methods/Analysis**, which has been slightly revised to provide a clearer definition (see **Pages 16**). A note has also been added to **Fig. 2 caption** for additional clarification. We feel that adding a box to the already crowded Fig. 2b should not be necessary with these clarifications.

8. *p8 , the authors mention the role that F315 plays – it seems to be responsible for a large change in diameter – does functional data exist for mutants at this position?*

Responses: Please kindly see responses to **Reviewer #1, points 8 & 12** above.

9. *Fig3. Needs to be clearer that the MD equilibrated structures shown are of the metal free state (I assume?)*

Responses: The caption has been updated to clearly state that the structures and properties shown are from simulations of the metal-free channel.

10. *p9 ...'represent true gating events'. The authors need to be careful in their choice of language – they observe dewetted states that are very likely to be non-conductive, 'true' is probably too strong a statement. Likewise, the next sentence is also incorrect – open dewetted pores have been seen before in other channels (see above).*

Responses: The wording “true” has been replaced with “highly plausible”. The subsequent discussion has also be completely revised to properly acknowledge that similar hydrophobic gating mechanisms have been predicted for several other channels (see **Page 10**).

11. *p10 In the discussion about mutations - in MthK, mutations in a similar region produce a change in single channel conductance, is this also the case for the A316 mutns. Could this provide any insight into the mechanisms involved?*

Responses: The A316 corresponds to A88 in MthK, which has a similar orientation in the open Mthk channel structure (PDB: 4hyo) and also locates near the narrowest point in the pore. It is possible that we are misunderstanding the comment above, but we have extensively discussed A316 mutations in the current work under the section “**Effects of mutations on channel activation correlate with the tendency of pore dewetting**”. We fully agree that these mutagenesis studies have provided key insights into BK gating mechanismq.

12. How stable is the A316D mutation – I assume no structural collapse occurs here but that it does in A316V – is there a correlation with movements and dewetting for these mutations – no RMSDs are shown (even for just this region). Introducing an Asp into this position may also cause repulsion and prevent collapse – did they try any other polar mutns at this position e.g. A316T or A316N?

Responses: All mutant simulations were stable (see a **new Fig. S11**). A316D does appear to expand the pore slightly and increase the volume (as reflected in a slight increase in the number pore waters, see **Fig. S10**). A316T and A316N have been characterized experimentally, which either decreases $V_{1/2}$ (A316T) or leads to constitutively open channel (A316N) (see **Fig. S9**).

13. p12. This is not the first description of a bona fide hydrophobic gate (see above).

Responses: We fully agree. Please see responses to point #2 above. Thanks.

14. methods – the choice of TIP3P water is appropriate for the FF used – but did they try any other water types?

Responses: We have performed an additional control simulation of the metal-free hBK channel using the Gromos G54a7 force field with SPC water model. The result, summarized in a **new Fig. S6b**, suggest that similar dewetting transitions can readily occur. The Method and result sections of the manuscript have also been revised to include the new simulation and result.

REVIEWERS' COMMENTS:

Reviewer #1 (Remarks to the Author):

The authors have revised the manuscript and responded fully to all of my previous comments.

I am happy to recommend acceptance of the revised manuscript.

Reviewer #2 (Remarks to the Author):

The paper has been significantly strengthened and the authors have adequately addressed many of the concerns that were raised.

One issue.

On p. 8 lines 4-5, it is stated that "fully and semi-hydrophobic model nanopores can adopt stable dry states when the radius is less than ~ 7 and ~ 5 Å, respectively". However, examination of the paper cited by the authors suggests that the radius should be less than 4.5 and 3 Å for fully and semi-hydrophobic model nanopores to adopt dry state, respectively. Furthermore, that analysis suggests that semi-hydrophobic pores such as in the pore of an ion channel can never be competently dry. As it stands, there appear to be discrepancies between the conclusions of the cited paper and the current paper that probably requires some comments.

Page 9, line 1: "with minimal" is repeated from the previous page.

Reviewer #3 (Remarks to the Author):

I am now very happy with the substantial improvements the authors have made to this manuscript which now provides compelling evidence for the role that hydrophobic gating may play in BK channel gating.

Responses to Reviewers' Comments (*all original comments shown in Italic*)

Reviewer #1 (Remarks to the Author):

The authors have revised the manuscript and responded fully to all of my previous comments. I am happy to recommend acceptance of the revised manuscript.

Responses: We greatly appreciate the reviewer's enthusiastic support!

Reviewer #2 (Remarks to the Author):

The paper has been significantly strengthened and the authors have adequately addressed many of the concerns that were raised.

One issue. On p. 8 lines 4-5, it is stated that "fully and semi-hydrophobic model nanopores can adopt stable dry states when the radius is less than ~7 and ~5 Å, respectively". However, examination of the paper cited by the authors suggests that the radius should be less than 4.5 and 3 Å for fully and semi-hydrophobic model nanopores to adopt dry state, respectively. Furthermore, that analysis suggests that semi-hydrophobic pores such as in the pore of an ion channel can never be competently dry. As it stands, there appear to be discrepancies between the conclusions of the cited paper and the current paper that probably requires some comments.

Responses: It appears the reviewer had mis-read Ref. 30, Aryal et JMB (2015). The authors clearly stated as part of the Figure 1 caption that "A fully hydrophilic pore remains fully occupied by water. However, a hydrophobic pore starts dewetting below 14 Å and becomes completely dewetted below ~ 8–10 Å. Semi-hydrophobic pores also exhibit similar dewetting below ~ 10 Å (vertical dotted line)." Note that the authors were referring to pore diameters in the discussion. **We have added the following comment to the main text:** "The observation that the pore of BK in the metal-free state can fully dehydrate suggests that the pore behaves more like the hydrophobic instead of semi-hydrophobic model pores."

Page 9, line 1: "with minimal" is repeated from the previous page.

Responses: We have double checked to make sure that the final version does not contain the repetition.

Reviewer #3 (Remarks to the Author):

I am now very happy with the substantial improvements the authors have made to this manuscript which now provides compelling evidence for the role that hydrophobic gating may play in BK channel gating.

Responses: We greatly appreciate the reviewer's enthusiastic support!